# Polymicrobial Infections and Biofilms: Clinical Significance and Eradication Strategies

**DOI:** 10.3390/antibiotics11121731

**Published:** 2022-12-01

**Authors:** V T Anju, Siddhardha Busi, Madangchanok Imchen, Ranjith Kumavath, Mahima S. Mohan, Simi Asma Salim, Pattnaik Subhaswaraj, Madhu Dyavaiah

**Affiliations:** 1Department of Biochemistry and Molecular Biology, School of Life Sciences, Pondicherry University, Puducherry 605014, India; 2Department of Microbiology, School of Life Sciences, Pondicherry University, Puducherry 605014, India; 3Department of Genomic Science, School of Biological Sciences, Central University of Kerala, Kerala 671316, India; 4Department of Biotechnology, School of Life Sciences, Pondicherry University, Puducherry 605014, India; 5Department of Biotechnology and Bioinformatics, Sambalpur University, Burla, Sambalpur 768019, India

**Keywords:** biofilms, polymicrobial, chronic infections, metagenomics, prevention

## Abstract

Biofilms are population of cells growing in a coordinated manner and exhibiting resistance towards hostile environments. The infections associated with biofilms are difficult to control owing to the chronicity of infections and the emergence of antibiotic resistance. Most microbial infections are contributed by polymicrobial or mixed species interactions, such as those observed in chronic wound infections, otitis media, dental caries, and cystic fibrosis. This review focuses on the polymicrobial interactions among bacterial-bacterial, bacterial-fungal, and fungal-fungal aggregations based on *in vitro* and *in vivo* models and different therapeutic interventions available for polymicrobial biofilms. Deciphering the mechanisms of polymicrobial interactions and microbial diversity in chronic infections is very helpful in anti-microbial research. Together, we have discussed the role of metagenomic approaches in studying polymicrobial biofilms. The outstanding progress made in polymicrobial research, especially the model systems and application of metagenomics for detecting, preventing, and controlling infections, are reviewed.

## 1. Introduction

Biofilms are a community of microorganisms protected within an extracellular matrix from external environmental factors, host immune system and antimicrobials, metabolic cooperation, and community-coordinated gene expression [1]. In biological systems, bacteria are often observed as biofilms rather than planktonic forms. Biofilms can be formed with single species, multispecies, or between different kingdoms. Microbial infections are mainly due to biofilm or by polymicrobial biofilm involving various bacteria, fungi or viruses. Mixed biofilm communities are involved in various environmental processes such as biodegradation, bioremediation, denitrification, etc. [2,3]. In wastewater, various oxygen-dependent nitrifiers and anaerobic denitrifiers coexist at different layers of biofilm [2]. There are five important stages in the establishment of mixed-species biofilms. The different phases of mixed-species biofilm formation include initial alterable attachment to the substratum, stable adherence, microcolony formation, maturation, and dispersal. The development and entablement of mature biofilms are highly dynamic and modulated by time, microbial interactions, and environmental signals [4].

The oral cavity of humans is one of the reservoirs of mixed-species biofilms. Oral microorganisms reside in the form of biofilms to survive in the dynamic nature of the oral cavity. The specific chemical signals produced by the oral commensals help in the formation of biofilms and can cause dental plaques in the oral cavity [5]. The oral pellicle, a proteinaceous layer that covers and protects the tooth, can also be an attachment substratum for forming biofilms [6]. The attachment of microbiota happens chronologically [7]. Initially, one species forms a layer on the tooth surface as an early colonizer, which serves as an attachment site for the next species through microbial interactions. This process is referred to as coaggregation, which occurs in two ways: (i) secondary colonizers bind to the molecules on the biofilm surface and continue the process of coaggregation, and (ii) the formation of aggregates by bacteria which causes a change in phenotypic character leading to further coaggregation process. Once the biofilms are formed, the microorganisms gradually modulate their gene and protein expression [7].

The interaction between microbes is complex and involves competition for space and nutrients. Due to the presence of high microbial loads in relatively less space, physical and chemical interactions developed gradually over the years of co-evolution [7]. The physiology and the function of the whole biofilm community often change during these interactions. Microbial activities are regulated by various interspecies interactions, such as metabolic interactions, quorum sensing, and antimicrobial compounds. Based on the type of interactions, bacterial species organize into three different spatial forms: interspecific segregation, co-aggregation, and stratification [8].

Some of the microbial interactions reviewed from the literature are cooperative interaction and antagonism, which affect the biofilm biomass, functionality, and tolerance when compared to mono-species biofilm [1]. Cooperative interaction facilitates the adhesion and growth and protects against antimicrobial agents among bacterial species. In Microbial Fuel Cells (MFC), multi-species develop biofilm over the surface of electrodes, which helps in the generation of electricity from organic compounds [9]. Synergy is defined as the greater effect that can be achieved when two or more species are combined together than by the individual species alone. The effect includes increased growth, antimicrobial resistance, virulence, and exopolysaccharide production [10]. Mutualism or synergism facilitates the exchange of metabolic products between species. In mutualism, there is a strong metabolic interdependence during intermixing but weak metabolic reciprocity between species, resulting in a spatial structure with initial segregation [11]. For instance, co-infection of *Pseudomonas aeruginosa* and *Staphylococcus aureus* was reported to delay wound healing (diabetic & chronic wounds) and to trigger host inflammation. Another type of interaction is called syntrophy, in which one species feeds on the byproduct of the other. *Fusobacterium nucleatum* and *Prevotella *intermedia** generate ammonia (metabolic byproduct), which increases the pH and creates an environment suitable for the growth of *Porphyromonas gingivalis*. It has been reported that the byproducts of *P. aeruginosa* protect *S. aureus* from aminoglycosides [12]. 

Antagonism or antibiosis is a competitive type of interaction where one species inhibits the growth of the other species. Organisms produce various secondary metabolites (interference competition), which inhibit or kill the competing species, through which they can avail available space, energy sources, and nutrients (exploitative competition) [13]. Studies on antagonistic relationships have revealed that they disrupt the biofilm architecture, which can be harnessed to develop treatment strategies against biofilm-associated diseases [10]. Due to the limited nutrient availability and space, the interspecific segregation among species will be higher in competitive or exploitative types of interactions.

Biofilm-based microbial infections have become a severe threat to public health. Based on the increasing cases of polymicrobial biofilm infections, this review article explores the recent updates on polymicrobial biofilm pathogens, interactions between and within microorganisms that exacerbate biofilm formation, and model systems to study polymicrobial biofilms. We have also explored the metagenomic approaches towards the surveillance of polymicrobial biofilms and the recent trends to mitigate polymicrobial biofilms, such as quorum sensing inhibitors, nanoparticle- and nanoconjugate-mediated therapy, antimicrobial photodynamic therapy (aPDT), phage therapy, combinatorial probiotics, etc.

## 2. Antimicrobial Resistance in Polymicrobial Biofilms

One of the most alarming effects of polymicrobial interaction is the development of antimicrobial resistance. The co-operation of different microbial communities within the biofilm renders them resistant to biocides. Several studies reported that the mixed microbial community is more resistant to disinfectants or antimicrobials when compared to mono-species biofilm. The antimicrobial resistance can be developed by different mechanisms in mixed populations [5] and are listed below. 

Composition of EPS matrix: Matrix supports the microbial cells for adherence, immobilization, and protects from environmental stress and antimicrobial agents. EPS composition varies from species to species and also with the environment. *P. aeruginosa* polysaccharide (Psl) is reported to provide resistance against colistin, polymyxin B, tobramycin, and ciprofloxacin, and a similar effect is also observed in non-psl producers such as *Escherichia coli* and *S. aureus,* possibly via electrostatic forces [14]. Matrix composition differs in multispecies biofilm, which confers more resistance than mono-species biofilm. *Candida albicans* protects *S. aureus* from vancomycin treatment by secreting exopolysaccharide, β-1,3-glucan, while *Streptococcus mutans* produces glucans that protect the *Candida* from fluconazole in mixed biofilms [15].Commensal-like interactions: One member in the community provides a suitable condition for the survival of other members in an inhospitable environment. It was exemplified by Elias and Banin, 2012 [5], who found that the presence of aerobes provides a better condition for the survival of anaerobes when the oxygen concentration is high.Alteration of physiology by neighbouring species: It was reported that HQNO (4-hydroxy-2-heptylquinoline-N-oxide) produced by *P. aeruginosa* could be used by *S. aureus* to increase the tolerance to antibiotics (vancomycin & tobramycin). Prolonged exposure to HQNO or with *P. aeruginosa* makes the resistant small-colony variants (SCV) of *S. aureus* [16].Resistance to host immune response: The alpha toxin of *S. aureus* disrupts the host immunity and the barriers of epithelial cells, which leads to the co-infection with *P. aeruginosa* and eventually causes pulmonary dysfunction [17].Interspecies genetic exchange: Horizontal gene transfer (HGT) of resistance genes in multispecies biofilm results in the emergence of resistance in evolutionarily distant species. HGT facilitates a range of adaptations, such as changes in metabolic levels, antimicrobial resistance, and biofilm formation. It was reported that the conjugative plasmid induces biofilm development and stimulates biofilm formation [8]. It was found that plasmid having the carbapenemase resistance gene (*bla*_NDM-1_) was transferred from *E. coli* to either *P. aeruginosa* or *Acinetobacter baumannii* via conjugation in multispecies biofilms. Moreover, exchange of mobile genetic elements such as *mecA* cassette has also been reported [12].β-lactamases-producing strains: β-lactamases are the enzymes that hydrolyse β-lactam antibiotics (cell wall-targeting drugs). Inactivation of β-lactam antibiotics effectively protects the cell itself and other cells in the polymicrobial biofilm. For example, co-culturing of *Haemophilus influenza* (β-lactamase producer) with *Streptococcus pneumoniae* (β-lactamase non-producer) increases MIC/MBC of amoxicillin (β-lactam antibiotic) [12].

## 3. Polymicrobial Infections

The colonisation of one microorganism can influence the colonisation of the other microorganisms in the host. In respiratory tract infections, viruses promote bacterial infection by suppressing the immune system, destroying epithelial cells, and upregulating the expression of molecules essential for the adhesion of bacteria. These viruses are also involved in middle-ear infections caused by bacteria, resulting in otitis media [18]. Polymicrobial biofilm infections affecting different parts of the body are depicted in Figure 1. Dental caries is the most common dental infection, affecting almost 60 to 80% of children and adults. The tooth surface serves as a site for the attachment of conditioning film (made up of salivary proteins and carbohydrates), to which the microorganisms form the biofilm [7]. Studies reported that approximately 48 h are required to transform individual cells into biofilms. *S. mutans*, the most common etiological agent causing dental caries, primarily attaches via glucosyltransferases and initiates the coaggregation of other bacteria, resulting in a polymicrobial biofilm. Such a type of biofilm prevents the host immune response by ceasing the host-signaling pathways [19]. 

Otitis Media or middle-ear infection affects the Eustachian tube located between the tympanic membrane and the inner ear. It is a childhood disease and rarely incident with the death of the person. When the infection is severe, it results in the loss of hearing ability of the patients. The bacteria which cause middle-ear infections are the commensals such as *S. pneumoniae*, *H. influenza*, *Moraxella catarrhalis*, which, in association with viruses such as influenza A virus, Adenovirus, Rhinovirus and Respiratory Syncytial Virus (RSV), form the infection. Monomicrobial infections caused by these microbial species have significant effects, but they also predispose the host to polymicrobial infections [7]. 

Lung infection, especially Cystic Fibrosis (CF), has been extensively studied for polymicrobial infection. It is an autosomal recessive disorder associated with microbial infection and leads to respiratory failure. Due to the lack of mucociliary clearance in CF patients, the inability to clear the pathogens trapped in the airways causes chronic polymicrobial infections. Pathogens such as *P. aeruginosa*, *S. aureus*, *Streptococcus milleri* group, *Burkholderia cepacia*, *Stenotrophomonas maltophilia*, *H. influenzae* and *C. albicans* cause infection in the airways. It was reported that the initial colonizers of lung infections in CF are *S. aureus* or *H. influenzae*, and the later colonizers are *P. aeruginosa* and finally *B. cepacia* [7,20]. Wound infections are formed when a patient’s epithelial barrier is compromised due to underlying conditions such as obesity and diabetic mellitus. Biofilm formed by the microbes delays the healing of the wound and renders it resistant to antibiotic treatment. Bacterial species involved in chronic wound infection are *P. aeruginosa*, *Enterococcus* sp., *S. aureus*, *Streptococcus* sp., and *E. coli.* Studies demonstrated that co-infection with *Bacteriodes fragilis* and *E. coli* showed polymicrobial interaction among both species, resulting in inflammation and pus-containing wounds [20]. Understanding the microbial species involved, predisposing factors of the disease progression, and the polymicrobial interaction between microorganisms are essential for diagnosing and developing treatment strategies. 

## 4. Polymicrobial Interactions

Microbial interactions play a vital part in maintaining microbiome structures. Mutualistic or commensal relationships among microbes develop into positive interactions, whereas parasitic or pathogenic interactions are negative. Novel techniques have helped to explore and characterize the microbial interactions occurring in the microbiome and enable the manipulation of the microbiome for better and improved medical treatments, environmental, and agricultural applications [21]. Most of the biofilms are observed as polymicrobial biofilm communities attached to inert or living surfaces. Polymicrobial biofilms are a group of various microorganisms, such as bacteria, fungi, and viruses, that live in a coordinated manner. In this, bacterial populations are embedded in the exopolysaccharide (EPS) matrix produced during their developmental stage [18]. The exopolysaccharide matrix consists of exopolysaccharides, extracellular DNA, lipids, and proteins. The EPS matrix is responsible for significant structural and functional properties of biofilms, including their survival and virulence [15].

Interestingly, bacterial biofilms formed during infections develop into polymicrobial interactions. The biofilm bacteria recruit other bacterial species to pursue polymicrobial interactions and regulate the gene pool provided by each of them. Thus, biofilm as a whole works synchronously to control and regulate the survival mechanisms of individual members. The requirements of biofilms, such as attachment to the target site, stimulation of host cellular senescence mechanisms to prevent the shedding of bacteria, and the production of plasma exudate for nutrition through local inflammation, are carried out by the gene pool of biofilm. The advantage of having polymicrobial groups in biofilm is that the genes contributed by each of the individual colonies perform their basic requirements [22,23,24]. The advantage of genomic plurality in polymicrobial biofilms is that it allows the development of novel strains and fosters persistent infections. The supragenome of some species in biofilms is shared among the other community members, developing its total gene pool. The genomic plurality is regulated by horizontal gene transfer occurring in the biofilm communities. The mechanisms other than the gene pool that enable their survival are passive resistance, quorum sensing pathways, metabolic co-operation, and by-product influence [12]. 

Polymicrobial interactions are generally observed on the host and environment surfaces. Though they are present in different body sites, the most studied and diverse polymicrobial interactions are from the oral cavity. The interactions among polymicrobial communities involved in pathogenesis are complex. The interactions may cause tight competition for nutrients and space or develop cooperative relationships to enable the growth of each partner in the polymicrobial colony [10]. The polymicrobial interactions can be synergistic, additive, or microbial interference mechanisms. In synergistic associations, establishing one microbe in a particular niche favors the host for the entry and colonization by one or more pathogens causing infections or disease [25]. More severe disease symptoms are often observed during synergy than the occurrence of conditions by the individual pathogens alone. For instance, studies showed wide microbial diversity during human periodontitis infections. The microbial diversity observed at the subgingival crevice indicated synergistic interactions. Likewise, highly dense microbial species were found in the lungs of CF patients associated with polymicrobial infections. The severity of lung infection is affected by the pathogens, *S. aureus*, *H. influenzae*, *P. aeruginosa*, and *B. cepacia* [26].

Additive polymicrobial interactions are observed within the biofilms attached to natural or artificial surfaces in the human system. Additive interactions lead to infections such as bacteremia, liver and soft tissue infections, otitis media, brain, lung, and abdominal abscesses by the combined effect of two or more non-pathogenic microorganisms [18]. The oral plaque biofilms are formed by several aerobic and anaerobic Gram-negative and positive bacteria and some *Candida* sp. They cause additive periodontal diseases. Actinomyces and *Streptococci* sp. initially colonize the tooth surface, and their interactions lead to the co-aggregation and co-adhesion of other pathogens [27]. 

In microbial interference, polymicrobial interactions between potent pathogens or probiotic organisms and pathogens may develop a niche unfavorable for colonization by other organisms [28]. For example, a flavivirus, GBV-C, resembles the hepatitis C virus but is not pathogenic in humans. GB virus C replication takes place in lymphocytes. The *in vitro* viral infections are associated with a decreased mortality rate in HIV-infected persons. The reduced mortality is due to the inability of HIV to replicate in the lymphocytes infected with GBV-C [28]. Antimicrobial synergism is observed with polymicrobial biofilms where the infectious agent shows higher antibiotic susceptibility than the individual species. For instance, in antimicrobial susceptibility testing, 60% of planktonic cells were inhibited with antibiotic combinations. In contrast, susceptibility towards antibiotics was observed in only 22% of bacteria in their biofilm state [29]. 

### 4.1. Bacterial-Bacterial Biofilms

Bacteria exhibit co-aggregation and co-localization mechanisms to interact with their partner species within a biofilm. Beneficial partner species with different genes can be selected within the biofilm by reversible co-aggregation methods. In contrast, during the co-localization process, favorable growth conditions are provided by beneficial bacteria to enable biofilm development [30]. The initial colonization of one species helps other partner bacteria. The initial colonizer prepares the substrate surface on which biofilm growth occurs and initiates the process of co-aggregation. The co-aggregation is mediated in two ways by the second colonizer. The second colonizer may bind to the specific surface molecules of the biofilms, or the group of bacteria co-ordinates among themselves and favors some phenotypic changes that lead to the co-aggregation on biofilms [7]. The co-aggregation process helps in the extended production of the biofilm matrix and colonizes the maximum surface area of the substrate, protecting from antibiotic stress [30]. One of the common polymicrobial biofilms is observed in acute otitis media, which is associated with 700 million cases reported every year globally [31]. Polymicrobial bacterial biofilms of otitis media, *M. catarrhalis*, non-typeable *H. influenzae*, and *S. pneumoniae* are responsible for the failed or ineffective antibiotic therapy [32]. A common polymicrobial interaction observed between *P. aeruginosa* and *S. aureus* in cystic fibrosis lungs and wound infections lead to enhanced severity in patients. It was observed that *S. aureus* infections are often experienced during childhood, whereas *P. aeruginosa* colonization increases along with age. *P. aeruginosa* has an antagonistic relationship with *S. aureus*. The coexistence and virulence of these versatile pathogens increase upon stress conditions such as antibiotic therapy [33].

### 4.2. Bacterial-Fungal Biofilms 

*Candida* species are one of the leading opportunistic fungal pathogens responsible for worldwide nosocomial infections. In general, *C. albicans* biofilms cause infections in immunodeficient patients. Recurrent candidaemia is one of the complications of candida infections, leading to more than 50% of the global mortality rate globally [34,35]. Complex multispecies interactions involving neutral, synergistic, and antagonistic associations are observed with *C. albicans* on both biotic and abiotic substrates [36]. The fungal-bacterial interactions occur through co-aggregation and adhesion. The adhesions, such as hyphal-associated adhesions of fungi and other adhesins present in the cell surface of bacteria, help in the interactions [37]. Fugal-bacterial biofilms are associated with bloodstream infections, and the treatment strategies focus on anti-fungal and anti-bacterial therapy. *C. albicans* forms polymicrobial biofilms along with *S. aureus* and *S. epidermidis* in nosocomial bloodstream infections and enhances the antibiotic resistance pattern [38]. The polymicrobial interactions of *Candida* sp. and *Streptococci* involved in oral candidiasis are synergistic. This association takes place with the help of adhesins present in the hyphal cell wall (Als 3) and bacterial cell surface (SspB). In this context, a bacterial partner assists the fungi to improve the biofilm formation and thereby pathogenesis of *Candida* by providing nutrients from salivary pellicle. Fungi enhance the Streptococcal growth by supplying nutrients and reducing the oxygen levels in the surrounding areas [39]. 

### 4.3. Fungal-Fungal Biofilms

Fungal-fungal mixed biofilm interactions are much less studied. The studies so far conducted have shown active polymicrobial synergistic interactions of fungal-fungal biofilms in infections. The mixed co-infection of *C. albicans* and *C. glabrata* showed synergy in developing denture stomatitis. *C. albicans* help *C. glabrata* to invade the host epithelium and to establish infection [40]. Other mixed fungal biofilms are between *C. albicans* and *C. dubliniensis*, and *C. albicans* and *C. rugosa*, which exhibited decreased susceptibility towards anti-fungal agents and enhanced infection rates in the host [41,42].

## 5. Clinically Relevant Human Polymicrobial Biofilm Infections

### 5.1. Oral Infections

Oral infections pose a significant global health threat owing to their high morbidity rate. Approximately 3.5 billion individuals are reported to live with uncured oral diseases. Most commonly observed infections are periodontitis, dental caries, stomatitis, and peri-implantitis [43]. Oral microbial dysbiosis may cause periodontitis, an inflammation of periodontal structures such as teeth, periodontal ligament, alveolar bone, and surrounding tissues. Periodontitis may lead to the loss of teeth if not treated properly. There is also a high risk of diabetes mellitus, adverse pregnancy outcomes, rheumatoid arthritis, cardiovascular complications, and respiratory infections in individuals with oral inflammation. The dysbiosis of the oral microbiome initially causes inflammation and later negatively impacts the host’s immune response [44]. Several microbial species reside and grow in the oral cavity, where more than 700 species have been observed. Among these, hundreds of species are involved in oral biofilm formation [45]. A deviation in the bacterial species of the subgingival region occurs during the incidence of inflammation. The pathogenic Gram-negative bacteria replace the symbiotic Gram-positive bacteria. Microbial complexes observed during the first stage of infection and disease progression are different. A group of anaerobic and pathogenic Gram-negative bacteria observed in the first stage of oral dysbiosis are called an orange complex and consist of *Prevotella nigrescens*, *P. intermedia*, *F. nucleatum*, and *P. micros*. As the disease progresses, the orange complex is replaced by the red complex, consisting of *Tannerella denticola*, *T. forsythia*, and *P. gingivalis* [46,47]. 

Orange and red complexes are the late colonizers of the oral tissues, whereas the early colonizers are yellow, green, and purple complexes. The early colonizers are from the genus *Actinomyces*, namely the blue complex. The yellow complex comprises species from the genus *Streptococcus*, the green complex species from the genus *Capnocytophaga* sp., *E. corrodens*, *A. actinomycetemcomitans* serotype A, and *Campylobacter*, and the purple contains *V. parvula* and *Actinomyces odontolyticus*. The colonization of the preceding complexes favors the occurrence of each complex during periodontitis. For instance, the red complex pathogens are mostly found to colonize most tissues in the presence of the red complex only. Red-complex pathogens are found to be dominant in the process of biofilm development during the final disease progression stages [48,49]. 

Generally, tooth decay and dental caries are associated with *S. mutans*, *Bifidobacterium* sp., and *Scardovia wiggsiae*. Metabolomics analysis has revealed that oral biofilms involved in polymicrobial interactions utilize carbohydrates and metabolize them into organic acids. The decreased pH and demineralization of dental tissues lead to the attachment of pathogens to the enamel and cause dental caries [50]. The acid produced by biofilms damage teeth enamel and lead to the cavitation and destruction of teeth [51]. Approximately 60–90% incidence of dental caries is observed in underprivileged children due to the acid produced by *S. mutans* biofilms. The virulence is modulated by the interaction of *S. mutans* with other bacteria. The spatial arrangement of species enables the production of protective barriers and improves their disease mechanisms or virulence. The unique three-dimensional architecture with a corona-like structure observed in the oral cavity is contributed by *S. mutans* as the center core surrounded by layers of other bacterial species [52].

In addition to bacterial biofilms, several fungal species also reside in the oral region. Oral candidiasis is a frequently occurring oral infection caused by *Candida* sp. [53]. *Candida* sp. is associated with multiple candidiasis and root caries in immunocompromised individuals. The ability of *Candida* sp. to change from yeast to hyphae growth enables their virulence towards the host. Other complications of *Candida* sp. are oropharyngeal candidiasis, denture stomatitis, where the soft and hard palate, buccal mucosal tissue, tongue, and mouth floor are infected [54]. Fungal species, which are commensal and non-pathogenic, can also contribute to the formation of biofilms by pathogenic species. For example, the physical and chemical signaling between *C. albicans* and *Streptococci gordonii* leads to the formation of dental plaques and can further help in the formation of biofilms [55]. Denture stomatitis is the inflammation of the oral mucosa and subsequent pathological responses linked to the denture surfaces near the tissue. Denture surfaces carry microbial biofilms and co-aggregate with *C. albicans*, enhancing the severity of infection. Polymicrobial biofilms of *P. gingivalis*, *A. actinomycetemcomitans*, *F. nucleatum*, *Lactobacillus* sp., and *Streptococcus* sp. are found along with *C. albicans* during denture stomatitis [56].

### 5.2. Wound Infections

Wound infections pose a significant global threat due to high morbidity and mortality rates. Infections of burns, surgical sites, and non-healing diabetic foot ulcers cause millions of deaths yearly. These are microbial contamination of wounds by either endogenous or exogenous sources [57,58]. Longer time environmental exposure of open wounds and a nutrient-rich wound bed create ideal conditions for the growth and multiplication of microbial pathogens [59,60]. The loss of skin integrity can expose sub-cutaneous tissue to colonization by microorganisms followed by multiplication. Microbial biofilms can easily develop on damaged tissues. Microbial composition of wounds includes cultivable as well as non-cultivable aerobic and anaerobic bacteria. The high polymicrobial load in wounds delays the wound closure and favors the emergence of antibiotic-resistant strains compared to the single-species biofilms [61,62]. The polymicrobial interactions in wounds help the partner species to establish and infect the tissues. Microorganisms colonize different niches in wound microenvironments to establish stable and persistent infections. Common symbiotic interactions observed in wounds are by *S. aureus* and *P. aeruginosa*. In a wound biopsy, *S. aureus* was found in the superficial areas, whereas *P. aeruginosa* was found in deeper areas and produces various virulence factors [63,64]. 

Wound biofilms act as mechanical barriers to the antimicrobial agents produced by the host and provided as prophylactic measures. In addition, the role of biofilms in hampering the re-epithelialization process, causing continuous inflammation and delayed wound healing, is well studied [65]. Chronic wounds are found to be severe, compared to acute wounds, where the treatment fails due to which skin integrity is not restored within 30 days [66]. Chronic wounds such as ulcers of a diabetic foot, decubitus, venous leg, and infections of a surgical-site are non-healing where bacterial interactions develop antibiotic-resistant biofilms [67]. It has been observed that less than 10% of acute wounds form biofilms, whereas chronic wounds are severe and approximately 60% of chronic wounds support biofilm growth [68,69].

Wound infections are mainly caused by the resident flora of neighbouring skin, and of oral and gut cavities. Chronic wounds, especially the surgical sites, are infected by the aerobic and endogenous pathogens *S. aureus*, *P. aeruginosa*, *E. coli*, *Enterococcus* sp., *Klebsiella* sp., *Enterobacter* sp., coagulase-negative *Staphylococci*, and *Candida* sp. Healing of soft tissue and bite wounds is delayed by the anaerobic organisms *Bacteroides fragilis*, *Clostridium perfringens*, *Prevotella* sp., *Porphyromonas* sp., and *Peptostreptococcus* sp. [70,71]. *S. aureus* and coagulase-negative Staphylococci are common in vascular, breast, cardiac, orthopaedic and ophthalmic surgeries and associated wound infections. Gram-negative bacilli and anaerobic organisms are associated with wounds of abdominal surgeries. The exogenous organisms are generally from operation theatres and the surrounding environment, including air, tools and materials used in surgery and personnel. Most common surgical-site infection outbreaks are due to infections by exogenous *S. aureus* and *S. pyogenes* [72]. Other than the multispecies interactions, inter-kingdom interactions involving fungi-bacteria in wounds have also been observed. The most commonly found fungal species are from the genus *Candida*, other than *Malessezia*, *Curvularia*, *Cladosporium*, *Trichophyton*, *Ulocladium*, *Engodontium*, and *Aureobasidium* [73]. Studies have shown a fungal-species presence of more than 50% in certain polymicrobial wound infections [74]. 

### 5.3. Diabetic Foot Ulcers

Diabetic foot ulcers are one of the complications associated with diabetic patients that increases the health care cost and mortality rate worldwide. Around 50% of microorganisms found in foot ulcers are pathogenic, which results in redness, purulence, swelling, warmth, pain, or induration in wounds and delays the wound healing process. The most predominant pathogen observed in diabetic foot ulcer is *S. aureus* along with other bacteria such as *P. aeruginosa* and *S. pyogenes* [75]. Other polymicrobial flora of foot ulcers associated with diabetes are *Enterococcus* sp., *Streptococcus* sp., *Acinetobacter* sp., *Corynebacterium* sp., *Porphyromonas* sp., *Prevotella* sp., and members of the family Enterobacteriaceae. Diabetic foot ulcers are found to be infected with normal foot skin flora, especially the anaerobes when the epithelial barrier is breached, which leads to the invasion of organisms. The abundance of anaerobes in wounds increases the severity and the wound-healing process. The chronic and recurrent nature of wound infection is due to the formation of polymicrobial biofilms and the expression of virulence phenotypes [76]. The infection of foot ulcers further cause deeper tissue infections, enhanced necrosis and amputation of lower limbs. Studies show that infection by drug-resistant microorganisms is the major cause of minor or major amputation and mortality rate even though ischemia and neuropathy play initial roles in the pathophysiology of diabetic foot ulcers [77]. Reports say that diabetic foot ulcers are responsible for around 50–70% of all limb amputations [78]. Moreover, there is an increased risk of death within 18 months from chronic infection [79]. 

Reports say that >265,000 deaths occur globally owing to burn wounds associated with biofilm infections. First-, second- and third-degree burns affect the epidermal, dermal and underlaying tissues of epidermal and bones, respectively. Bacterial pathogen colonizes the burn sites according to the degree and intensity of the burns. The more severe the burns, the greater the probability that pathogens can invade the circulatory system, causing systemic infections, sepsis and bacteremia. The region surrounding a burn is rich in wound exudates. The burn exudates act as good niche for the establishment of pathogenicity in the host. Biofilms of nosocomial pathogens, *S. aureus*, *P. aeruginosa* and *A. baumannii* prevent the control of burn wounds [80]. The polymicrobial biofilms of burn wounds are initiated by Gram-positive bacterial infection, as they are able to withstand thermal stress, followed by Gram-negative bacteria. The interaction of *S. aureus* and *P. aeruginosa* is very severe, as they enhance the virulence and pathogenicity through quorum sensing. The increased inflammation by these pathogens is marked with the release of different cytokines [81,82,83]. The interactions of *A. baumanni*i and *P. aeruginosa* are observed in battle-associated burn wound infections, where their impact on wounds is less than that of other bacterial biofilms [84]. 

### 5.4. Respiratory Infections

The polymicrobial communities involved in respiratory tract infections are associated mostly with the CF patients. Cystic fibrosis is a genetic disease affecting 70,000 people globally. It results from a dysfunction or mutation in the cystic fibrosis transmembrane conductance regulator (CFTR) gene. As a result of mutation, cystic fibrosis airways are filled with nutrient-rich and viscous mucus secretions [85]. These hinder mucociliary clearance and lead to the invasion of pathogens, causing chronic infections. It has been observed that the cystic fibrosis airways of patients are colonized by diverse and dynamic arrays of microbial communities. The polymicrobial communities of CF patients communicate through quorum sensing mechanisms, secretion and recognition of small metabolites and cell surface proteins [85,86]. The microbial interactions lead to chronic infections, which generate enhanced inflammatory responses, damage to the airway tissue, declining lung functions and early death [87]. 

The lungs of CF patients comprise polymicrobial communities of bacteria and fungi. They are from the bacterial genera *Streptococcus*, *Veillonella*, *Actinomyces*, *Prevotella*, and *Rothia*, sp. *P. aeruginosa*, *B. cepacia*, *S. aureus*, *S. maltophilia*, *Streptococcus milleri*, *H. influenzae* and the fungi *Aspergillus fumigatus* and *C. albicans*. The lung microbial communities vary with age. For instance, higher microbial diversity is observed in younger patients, whereas older patients have lower diversity [7,88]. Among the key pathogens leading to chronic infections, *P*. *aeruginosa* and *S*. *aureus* are the most studied pathogens in the CF lung environment. These two pathogens colonize the airways, affect lung functions, and interfere with anti-bacterial therapy by adapting to various metabolic interactions [89]. Thus, antibiotic therapies targeting a single bacterial pathogen cannot effectively treat CF lung infections. 

### 5.5. Otitis Media

Otitis media is the inflammation of the middle ear, and it occurs frequently in child populations. During acute otitis media, middle ear fluid or effusion is observed along with other symptoms such as fever, irritation, otalgia and otorrhea. Sometimes middle ear infections are observed only with the presence of effusion [90]. Although the mortality rate associated with otitis media is low, there is a high morbidity rate and economic burden. The complications associated with otitis media in children are hearing loss, and delays in education, behavioural and language development processes. Globally, around 65 to 330 million cases are reported for otitis media, out of which 60% of cases are associated with hearing loss [91,92,93]. The otopathogens involved in the monomicrobial or polymicrobial biofilms of otitis media are non-typeable *H. influenzae*, *S. pneumoniae*, and *M. catarrhalis*. These pathogens are associated with 95% of the reported cases of otitis media [94]. Usually, the upper respiratory tract viral infections can initiate the bacterial infections of middle ear. The most common upper respiratory viruses associated with the incidence of otitis media are influenza virus A and B, Adeno virus, Enterovirus, Coronavirus, RSV, Rhinovirus, and Parainfluenza virus 1, 2 and 3 [95,96]. MRSA and *P. aeruginosa* interactions in the upper respiratory tract also lead to many chronic infections, including chronic otitis media [97]. 

## 6. Model Systems to Study Polymicrobial Biofilms

There are several model systems developed to study the interactions of polymicrobial biofilms and associated infections. *In vitro* and *in vivo* model systems are employed to study polymicrobial infections caused by mixed biofilms and to evaluate antimicrobial susceptibility testing. There is growing evidence showing the developments in model systems used to study polymicrobial infections [98]. The *in vitro* models include cell-line infection systems, whereas *in vivo* animal models include rats, mice, etc. Appropriate animal models are essential to understand the complex nature of polymicrobial infections and their treatment methods. The merits of using animals over *in vitro* models include the presence of an excellent immune system, well-arranged organ systems, availability of numerous molecular reagents to study immune responses, and the genetic background of inbred mouse to decipher the pathogenesis of polymicrobial communities [99]. Table 1 represents different model systems used to study polymicrobial biofilms.

These *in vitro* model systems are cheap, reproducible, and have fewer ethical concerns in studying composition, biofilm formation, and in evaluating anti-microbial testing for polymicrobial communities. There are *in vitro* static (microtiter plate) and continuous chemostat models to study oral biofilms. The continuous models include chemostat (bioreactor with continuous flow of media and nutrients), flow cell (slides comprising media) and constant-depth film fermenter (fermenter containing coupons suspended from the lid). The modern technology including microfluidics, continuous and static models allow the growth of biofilms [100]. For instance, a costar 24-well flat bottom cell culture plate was employed by Manavathu and co-workers to develop a polymicrobial biofilm model of *A*. *fumigatus* and *P. aeruginosa*. These pathogenic biofilms are observed in the CF lung airways, where they complicate antibiotic treatment. Thus, the *in vitro* biofilm static model enabled studies of the effect of antibiotics and combinations thereof (cefepime, tobramycin and posaconazole) [101].

*In vitro* models allowed studies of the growth of aerobic and anaerobic bacteria in chronic wounds. Chronic wound biofilms comprise mixtures of aerobic and facultative aerobic pathogens such as *P. aeruginosa*, *E. faecalis,* and *S. aureus*. Lubbock chronic wound biofilm (LCWB) model was used to evaluate the growth of facultative anaerobes in a completely oxygen-rich environment of wounds. The LCWB model provides similar circumstances for *in vivo* wound biofilms. The *in vitro* model is enriched with wound-simulating media, oxygen content, nutrients and an insoluble fibrin network provided by *S. aureus*. This wound model enables the attachment of bacteria and to form biofilms [110]. LCWB is one of the widely used models for studying the effectiveness of many anti-microbial agents. For the *in vitro* evaluation of anti-microbial dressings, a 48-h viable LCWB wound biofilm model was employed. The LCWB model was transferred into an artificial wound bed containing gelatin and agarose which remained viable for next 48 h and served as an effective pre-clinical *in vitro* experiment to study polymicrobial biofilms of *S. aureus*, *E. faecalis*, *B. subtilis*, and *P. aeruginosa* [111].

*Caenorhabditis elegans* is a transparent nematode, another widely used model system due to the ease in maintenance to study molecular and genomic approaches in microbial infections [112]. It serves as an excellent *in vivo* model to decipher host-pathogen interactions. This model allows the analysis of various virulence factors of pathogens and to study host defence molecules produced upon infection. The pathogenic biofilms initially colonize in the gut, leading to the disruption of the gut lining and eventually organ dysfunction and death [113]. A research group evaluated the role of EPS in the formation of polymicrobial biofilms of *S. epidermidis* and *C. albicans* using a *C. elegans* infection model. The study showed that hyphal formation by fungi and EPS production of bacteria led to increased virulence in the infected nematode and eventually caused death [114].

*In vivo* vertebrate models are most reliable and promising approaches to study polymicrobial biofilms and their interactions with the host. Dalton and colleagues developed a murine wound infection model to study polymicrobial interactions of wound pathogens *S. aureus*, *P. aeruginosa*, *E. faecalis*, and *F. magna* [57]. The mouse wound model displayed the role of polymicrobial biofilms in delaying wound closure and enhancing antimicrobial resistance. It was observed that polymicrobial wound infections showed improved antibiotic tolerance and delayed wound healing compared to monospecies biofilm interactions [57]. Polymicrobial biofilm formation by *P. aeruginosa* and *B. cenocepacia* in CF lungs were demonstrated in CFTIR-deficient mice. The co-infection of bacterial pathogens in mice led to the establishment of polymicrobial biofilms followed by chronic lung infection. The authors studied the effect of interspecies interactions on the development of chronic infections and enhanced production of inflammatory responses in the mice model [115].

The polymicrobial consortium of periodontitis consisting of *P. gingivalis*, *T. denticola*, and *T. forsythia* and its synergistic interactions leading to the increased inflammation was studied using a rat model. In the polymicrobial disease model, rats infected with pathogens exhibited chronic inflammation of periodontal structures, causing alveolar bone resorption. Thus, the rat model enabled an evaluation of the combined virulence mechanism of periodontal pathogens leading to chronic inflammation [116]. Likewise, a pneumonia mouse model with polymicrobial infections caused by *P. gingivalis* and *T. denticola* led to enhanced respiratory infections. The polymicrobial infections showed a higher mortality rate in the mouse model compared to the monospecies infections. The pathogenic biofilms developed bronchopneumonia and lung abscesses in mice which caused a higher mortality rate and markedly higher production of inflammatory cytokines [117]. The otitis media rat model was used to study the interactions of MRSA and *P. aeruginosa* in a host. The polymicrobial biofilms were established in the middle ear, causing chronic suppurative otitis media. The colonization of bacteria in the middle ear of rats caused increased production of inflammatory responses that were observed during gene expression studies [118]. Thus, the model systems are highly essential for evaluating the polymicrobial interactions and establishing biofilm infections and host-pathogen responses.

## 7. Metagenomic Approaches in Detection, Prevention, and Inhibition of Polymicrobial Biofilms

The current standard approach for polymicrobial detection remains culture-dependent [119]. However, culture-dependent approaches are prone to disadvantages, such as low turnaround and false negatives for slow-growing or low-titer pathogens. Such setbacks can be circumvented by implementing novel molecular techniques such as metagenomics. The advantage of metagenomics is mainly due to its culture-independent and high-throughput nature. Due to its high-throughput nature, metagenomics is widely used in exploring the entire microbial community of environmental or host microbiomes. Metagenomic sequencing can be loosely classified into shotgun or amplicon metagenomics. Although shotgun NGS is the gold standard for metagenomic analysis, 16s rRNA amplicon sequencing is also commonly used. The sensitivity of metagenomic techniques in detecting pathogens in clinical samples has also been demonstrated in several studies. Its advantages over the standard clinical approach include its sensitivity, low sample requirements, identification of pathogens not commonly observed in the standard clinical approach, fast turnaround, and potential to detect novel approaches for infectious disease treatments. The shotgun metagenome approach has several advantages over 16s rRNA metagenomics. It can provide information on the antibiotic resistance mechanism, mutations, metabolic potential, species/strain level information on the taxonomic diversity, and whole genome reconstructions [120]. On the other hand, 16s rRNA metagenomic is cheaper in terms of cost per sample (Figure 2).

The human microbiome comprises an intricate microbial network, such as the oral microbiome comprising a complex acidogenic and aciduric microbial community of >700 microbial species, several of which are yet unculturable. Among them, the Lactobacilli and mutans Streptococci (i.e., *S. mutans* and *S. sobrinus*) are known to cause caries [121]. The complex interaction between the microbial community has led to the polymicrobial synergy and dysbiosis (PSD) hypothesis, which states that oral disease could be the consequence of dysbiotic microbials rather than a specific pathogen. Similarly, the keystone pathogen hypothesis states that certain low-abundancy pathogens can modulate a healthy microbiome into a dysbiotic microbiome. Polymicrobial natural infections can be viewed based on the interactions between the species. Microbial interactions are often evolutionarily related, and the species within the polymicrobial community interact to protect each other from antibiotics [122]. Metagenomic analysis of such complex microbial communities has provided novel insights into the polymicrobial infections and involvement of keystone pathogens [123]. For instance, millions of diabetic patients are affected with a diabetic foot, characterized by bacterial infections and biofilm. Metagenomic analysis of the diabetic foot identified enrichment of multiple pathogens, including *Streptococcus* and *Corynebacterium*, while traditional culturable techniques identified *Pseudomonas*, *Proteus*, *Enterococcus*, and *Staphylococcus* [124]. Similarly, metagenomic studies of tuberculomas (*n* = 14) have revealed the lack of *M. tuberculosis* as dominant taxa in most samples, suggesting the polymicrobial nature [125]. Thus, metagenomic studies have provided novel insights into polymicrobial infection and their interactions mediated by keystone pathogens.

The sensitivity of metagenomics has also been demonstrated in the detection of polymicrobial infections in arthroplasty, surgical restoration of joint functions, which is mainly complicated by prosthetic joint infection (PJI). A routine clinical approach for the detection of PJI yields 20–35% false negatives [126]. 16s rRNA amplicon metagenomic screening of PJI though synovial fluid (SF) (*n* = 22) of PJI patients (*n* = 11) identified every pathogen that was detected with traditional cultures and also categorized them based on the arthroplasty stages and reduced turnaround time (two days instead of seven days in standard clinical approach). Similarly, the application of 16s rRNA metagenomics in polymicrobial detection has been studied in the sputum of cystic fibrosis (CF) patients with promising results: metagenomics was able to detect a significantly higher number of unique microbes (*n* = 122), including low-abundance and fastidious microbes compared to the culturable method (*n* = 18) [127]. Furthermore, the sputum samples could be classified into multiple groups (*n* = 5), suggesting the implication of a more specific diagnosis. The higher detection rate of pathogens through metagenomics has also been demonstrated through 16s rRNA amplicon metagenomic analysis of brain abscesses (*n* = 51) which detected more bacterial taxa when compared to the cultural approach with at least two bacterial taxa detected in most of the samples (*n* = 31), suggesting the polymicrobial nature of the disease [128]. Metagenomic analysis of samples with monomicrobial infections (*n* = 8) has a detection rate of 100%, but only about 58.2% and 74.5% at species and genus level, respectively, for polymicrobial infections (*n* = 55) [129]. Metagenomic analysis was also more rapid, with less than 24 h turnaround, and cost-effective compared to the standard clinical approach. The metagenomic approach has also been reported to detect potential pathogenic bacteria that were not detected in the culturable approach and predicted antibiotic susceptibility for 76.5% in the polymicrobial samples [129]. The strength of metagenomics has also been demonstrated in the detection of community-acquired pneumonia (CAP) (*n* = 59), which was able to detect pathogens not detectable with traditional techniques; some of the patients were detected with polymicrobial infections (*n* = 31), which led to increased outcomes and reduced the mortality rate [130].

Current standard practice for sepsis diagnosis is based on the culture of microbes from the bloodstream. However, the procedure is prone to underestimation or false negatives when the causative agents are in low abundance or difficult to grow. Sepsis can also be the manifestation of the polymicrobial interaction. A recent metagenomic study fortified with machine learning on 287 cohorts demonstrated that polymicrobial sepsis infections in blood-borne infections could be efficiently detected [131]. The authors report interesting findings. The model without the culture-confirmed pathogens performed well, and the model with only single-feature did not perform well, suggesting the polymicrobial nature of the species. Shotgun metagenomic analysis has also been highly efficient in detecting pulmonary infection. For instance, a large-scale metagenomic analysis of patients (*n* = 235) with suspected pulmonary infections or in a respiratory intensive care unit (RICU) or ventilator identified lower alpha diversity in confirmed patients [132]. A shotgun metagenomic study of periodontitis patients (*n* = 43) revealed that healthy subjects had higher alpha diversity, and the development of a machine learning model with a naive classifier was able to classify patients and healthy subjects with an accuracy of 94.4% [123]. The authors also determined multiple antibiotic resistance genes and polymicrobial keystone species: *P. gingivalis*, *Haemophilus haemolyticus*, *Prevotella melaninogenica*, and *Capnocytophaga ochracea*. Such studies provide clear evidence that the implementation of machine learning in metagenomic analysis has promising potential in disease detection and specialized therapeutics based on the nature of the infection.

One of the challenges in the shotgun metagenomic is the low sequencing depth of bacterial DNA due to the high load of host DNA. In the shotgun metagenome of human biopsy samples, most reads (>95%) can be attributed to the human host. Hence, a large number of reads are not suitable for the diagnosis. A novel DNA extraction technique was introduced wherein the intact DNA from extracellular or intact human cells is removed: the human cells are lysed in hypotonic treatment followed by digestion of the extracellular DNA with endonuclease digestion [133]. With such selective removal of the host DNA, the resultant reads are substantially high bacterial DNA reads and consequently detect more taxa. Furthermore, despite the sensitivity of the metagenomic approach, some reports have indicated the weakness of metagenomics in comprehensively detecting pathogens [134]. Hence, it is also important to note that some of the microbes detected through the cultivable approach were not detected in the NGS approach. However, the authors note that the de-nosing step in the metagenomic pipeline should be performed with care since closely related sequences are grouped as one, which can lead to loss of genetic diversity in the sample [127]. Hence, metagenomics can be applied in clinical settings as a supplementary approach to the existing cultivable approach. In addition, future directions for polymicrobial detection through metagenomics would include (i) implication of longer reads to obtain longer reliable assembled reads with higher resolution of identification, (ii) reduction of the cost per GB affordable to the clinics in low and middle-income countries, (iii) automatic pipelines friendly to the medical practitioners, (iv) open-source machine learning models in determining infection sub-groups, (v) application of machine learning approaches in denoising, quality control steps, and suggestion of the suitable measures based on the microbiome profiles.

### Discovery of Novel Biofilm and Quorum Sensing Inhibitors through Metagenomics

Biofilm formation by pathogens is one of the leading causes of antibiotic resistance and protects the pathogens from the host immune system. Thus, biofilm has rendered most antibiotics ineffective. Inhibition of biofilm is one of the main approaches to address microbial infection since ~75% of the microbial pathogens produce biofilms. Quorum sensing (QS) modulates microbial virulence, biofilm, and overall community behaviour. Hence, quorum sensing inhibitors (QSI) are potential agents for modulating biofilm formation and the virulence of the pathogens. N-acyl-homoserine lactones (N-AHLs) are a common class of QS signaling molecules in bacterial cell-cell communication. N-AHLs can be inactivated, also known as quorum quenching (QQ), by enzymes such as N-AHL hydrolase and N-AHL acylases. Recently, quorum quenching activity of oxidoreductases was identified from a soil metagenome [135]. It significantly reduced biofilm formation, pyocyanin production, motility, and the transcription of *lasI* and *rhlI* in *P. aeruginosa*, suggesting its importance in QS and biofilm inhibition strategies.

Since metagenomics does not rely on selective enrichment of genes or enrichment of the microbial community, it has the potential to discover novel QQ enzymes. The strength of metagenomics in discovering novel enzymes with QQ activity has been demonstrated in several reports. Novel QSIs such as bpiB01, bpiB04, bpiB07, and aii810 were discovered through a metagenomic approach that encodes for N-acyl-homoserine lactonase and exhibits antibiofilm activity in *P. aeruginosa* [136,137]. It was stable at below 40 °C, with neutral pH, and attenuated biofilm formation and the virulence of *P. aeruginosa*. Similarly, a novel hydrolase, 70 kDa BpiB05, that acts on N-AHLs, was identified through metagenomics that is dependent on Ca^2+^ reducing *P. aeruginosa* motility, pyocyanin, and biofilm [138]. It has been proposed that marine microbes are a rich source of bioactive molecules with QSI activity [139]. Screening of a metagenomic library constructed from a marine microbial community exhibited a strong QSI activity by ~7% of the clones (out of 2500 clones). The clones were able to disrupt and reduce the QS and biofilm formation in *P. aeruginosa* and *A. baumannii* [140].

Small peptides are a potential agents for biofilm inhibition. However, experimental screening of thousands of peptides is costly and resource-intensive due to the limited number of peptides with antibiofilm properties. In this context, prediction models based on the Support Vector Machine (SVM) approach have been attempted, with promising results [141]. Similarly, efforts to extract bioactive peptides from metagenomes also show great potential. Features from the primary structure, physicochemical features, and NMR spectra are fed into various algorithms such as random forest and SVM [142].

## 8. Innovative Approaches to Mitigate Polymicrobial Biofilms

There are several methods or agents that inhibit the formation of biofilms by targeting different biofilm developmental stages. Several novel methods or anti-biofilm agents that are under development can serve as alternatives to antibiotic treatment to reduce the emergence of antibiotic-resistant strains in future [143] (Figure 3; Table 2). As biofilms are highly resistant to antibiotics and fail to reach the deeper layers of biofilms, alternative methods which can degrade biofilm matrix and destroy the deeper resistant cells are recommended [144]. Antimicrobial resistance is often observed in polymicrobial biofilms owing to the presence of antibiotic-resistance genes, interspecies genetic exchange/transfer, production of microbial metabolites and quorum sensing signaling, leading to the altered antibiotic susceptibility. Therefore, multiple mechanisms are adopted by polymicrobial biofilms to promote antibiotic resistance [145]. For instance, transfer or exchange of antibiotic-resistant genes among polymicrobial communities often favours the enhanced antibiotic resistance. There is evidence of horizontal gene transfer (HGT) among mixed cultures of methicillin-resistant *S. aureus* (MRSA) and *E. faecium*. In mixed cultures, the presence of vancomycin (*vanA*) and tetracycline (*tetU*) resistance genes were detected in MRSA, yet had not been present previously [146]. Conjugation and/or transformation are also involved in the transfer of resistance genes in polymicrobial cultures. In a study, the transfer of tetracycline (*tetM*)-resistant genes to *Streptococcus* sp. was observed either by conjugation or transformation with *Veillonella dispar* [147].

Studies showed polymicrobial biofilms of *S. aureus* and *C. albicans* are more recalcitrant to antibiotic treatment regimes. In particular, the polymicrobial biofilms showed 10 to 100-fold resistance to rifampicin, ciprofloxacin, vancomycin, oxacillin and delafloxacin in comparison to the monotherapy in sessile *S. aureus* cells. The increased antibiotic resistance profile is correlated to the formation of nutrient-deficient persister cells in the biofilms [148].

There are enzymes or factors encoded by the bacteria to inactivate or degrade antibiotics. One such example is beta-lactamases, produced by a group of bacteria to inactivate cell wall synthesis inhibiting β-lactam antibiotics. It has been observed that these enzymes not only help their own bacterium which produces the enzyme but also protects the neighbouring bacteria in the polymicrobial interactions [149,150]. Beta-lactamases produced by otopathogen *M. catarrhalis* confer resistance to beta lactam drugs; other pathogens that produce beta-lactamases are *H. influenzae* and *S. pneumoniae* [145,151]. There are several reports of polymicrobial biofilms of bacteria and fungi which showed decreased vulnerability towards different generations of antibiotics. The growing challenge of antibiotic resistance in polymicrobial biofilms causing chronic and clinical infections recommends alternative and effective therapeutic modalities. New anti-biofilm molecules, compounds targeting interspecies interactions/quorum sensing, and anti-biofilm agents combined with commonly available antibiotics are under investigation and will be available in the near future [152].

### 8.1. Nanoparticle and Nanoconjugate Mediated Therapy

Nanotechnology deals with particles of 10 to 100 nm size widely used in different fields such as medicine and dentistry. The unique features of nanoparticles enable them to be used as effective and alternative agents in the treatment of infectious diseases and as drug delivery agents. They have a small size, high surface to volume ratio, chemical and biological reactivity [167]. There are metallic nanoparticles, polymer nanoparticles and nanoparticles with antimicrobial coatings used effectively in the management of infections. Metallic nanoparticles have the ability to interact with microbial membranes, nucleic acids, and proteins, resulting in the anti-microbial activity [168].

Nanoparticles exhibit enhanced bioavailability and targeted delivery of drugs to biofilms. There are effective drug delivery nanoparticles made of lipids, silica and polymers. These delivery agents can safely deliver drugs to the target site without being affected by bacterial-deactivating enzymes such as β-lactamases [169]. Metallic nanoparticles such as silver, gold, iron, copper, etc., can act as anti-biofilm agents [170]. Silver nanoparticles (AgNPs) are widely explored anti-bacterial and anti-fungal agents. Yasinta et al., 2021 studied the ability of AgNPs to inhibit polymicrobial biofilms of *E. coli* and *C. albicans*. The synthesized silver nanoparticles entered the EPS matrix and degraded the extracellular matrix growth, leading to the destruction of polymicrobial biofilms. Thus, AgNPs showed dual action by degrading EPS matrix and killing the polymicrobial biofilms of *E. coli* and *C. albicans* [171].

Positively charged silver nanoparticles inhibited single-species biofilms and polymicrobial biofilms of MRSA and *C. albicans*. AgNPs inhibited the MRSA/*C. albicans* biofilms in a dose-dependent manner, substantiating the role in inhibition of most of hospital-acquired infections by these biofilms. MRSA/*C. albicans* biofilms are found in indwelling devices, causing blood-stream infections. In addition, authors explored AgNPs coated catheters to prevent the formation of polymicrobial biofilms and provided evidence of charged silver nanoparticles and nanoparticle-coated catheters in the inhibition of polymicrobial biofilms of MRSA/*C. albicans* [154]. Along with silver, gold nanoparticles (AuNPs) were also explored as anti-biofilm agents. Interestingly, hybrid nanoparticles were prepared using silver and gold to evaluate their efficacy towards polymicrobial biofilms. Hybrid nanoparticles at very low concentrations inhibited the biofilm formation in *P. aeruginosa* and *S. aureus*. Moreover, the mechanism of action towards polymicrobial biofilm was found to be the production of intracellular reactive oxygen species (ROS) [172].

Curcumin is one of the phytochemicals which showed an anti-biofilm effect against a broad spectrum of pathogens. Researchers studied the role of curcumin-loaded chitosan nanoparticles against monospecies and polymicrobial biofilms of *C. albicans* and *S. aureus*. Chitosan nanoparticles work as a good drug delivery agent for hydrophobic curcumin to the biofilm matrix and favours sustained delivery of drug. Thus, nanoparticles improve the therapeutic effect of curcumin when loaded onto chitosan nanoparticles, which was confirmed by a reduced ability to form polymicrobial biofilms. Curcumin-loaded nanoparticles reduced the biofilm matrix thickness and favoured microbial death [153]. Maxillofacial silicone prostheses are employed in the therapy of head and neck defects. Monospecies or mixed biofilms are one of the reasons contributing to the degradation of these prostheses. Thus, silicone prostheses coated with silver nanoparticles inhibited the bacterial-fungal biofilm formation *in vitro*. The release of silver ions and cellular permeabilization led to the inhibition of polymicrobial biofilm formation by *S. aureus* and *C. albicans* [173]. Thus, the studies explored the opportunities of using nanosized particles to treat polymicrobial biofilms and to control the emergence of antibiotic-resistant strains.

### 8.2. Antimicrobial Photodynamic Therapy (aPDT)

Antimicrobial photodynamic therapy is an emerging and effective therapeutic strategy to treat acute and chronic infections caused by planktonic cells and biofilm pathogens. aPDT is a light-dependent treatment method which involves the production of reactive oxygen species through type 1 or type 2 pathways [174]. In aPDT, a non-toxic and light-sensitive compound termed a photosensitiser (PS) in its excited state interacts with molecular oxygen surrounding the cells and produces ROS such as hydroxyl free radicals, peroxides, superoxides and singlet oxygen [175]. ROS can act on different cellular targets such as proteins, DNA and lipids, thus potentiating the non-specific action of aPDT towards the elimination of resistant strains [176]. The role of PDT in the treatment of broad-spectrum biofilm pathogens was studied in both *in vitro* and *in vivo* conditions. Several studies are in progress with pre-clinical and clinical trials to evaluate the effectiveness of aPDT [177,178,179]. The mechanism of action of aPDT against polymicrobial biofilms is provided in the Figure 4.

Polymicrobial Gram-positive and Gram-negative pathogens are associated with the failure of endodontic treatments. Methylene blue (MB) is one of the widely used anti-microbials (PS) and showed effective phototoxicity against polymicrobial pathogens causing endodontic infections. Methylene blue (10 µg/mL) mediated aPDT reduced polymicrobial biofilms comprising *Actinomyces israelii*, *F. nucleatum* subspecies *nucleatum*, *P. gingivalis*, and *P. intermedia* on root canals. Thus, aPDT is an excellent alternative for the treatment of root canal infections [180]. Toluidine blue O (TBO) is an excellent anti-bacterial cationic PS used for broad-spectrum activity [181]. Akhtar et al. 2021 employed a novel nano-phototheranostic approach to prevent polymicrobial and monomicrobial biofilms of diabetic foot ulcer. In this method, anti-microbial agents, silver and gold nanoparticle cores, were coated with chitosan and conjugated with TBO to enhance the overall aPDT mechanism. The synthesized and photoactivated nano-PS conjugate reduced both monomicrobial and polymicrobial biofilms containing two versatile diabetic foot ulcer-causing pathogens, *P. aeruginosa* and *S. aureus* [182]. Similarly, nanoparticles were conjugated with TBO for enhanced loading and maximum phototoxicity against wound polymicrobial biofilms. Anionic surfactant dioctyl sodium sulfosuccinate-alginate nanoparticles were prepared and loaded with TBO and used against planktonic and mixed biofilms of MRSA and *P. aeruginosa*. Authors observed more reduction of biofilms in the presence of photoactivated nano-PS conjugates compared to the free TBO. Thus, the study potentiated the use of nanoparticles to enhance the efficacy of PDT and to reduce antibiotic-resistant chronic pathogens [183].

aPDT was found to be effective in the eradication of otopathogens involved in middle ear infections such as *H. influenzae*, *S. pneumoniae* and *M. catarrhalis*. In a previous study, Chlorin e6 (Ce6) was confirmed as effective PS against sessile cells and monomicrobial biofilms of otopathogens [164]. A recent work from same group reported that Ce6 can equally be effective against the polymicrobial biofilms of otopathogens at lower concentrations and optimized PDT parameters. Thus, the work suggests aPDT for the effective treatment of recurrent and chronic otitis media with a decreased rate of antibiotic resistance [94].

### 8.3. Antimicrobial Peptides (AMPs)

Interest in antimicrobial peptides as anti-biofilm agents has increased dramatically in the past few years. These are small molecules observed in almost all life forms, such as multicellular organisms to bacteria. Antimicrobial peptides are involved in the primary defense mechanism of innate immunity where they attack invading pathogens [184]. There are different groups of AMPs based on their solubility, net charge, secondary structure, amphipathicity, and hydrophobicity. AMPs acts either on the cell membrane, causing membrane perturbation, or on intracellular targets such as DNA, proteins and cell walls. Other than the general mechanism of action, AMPs can interfere with the polysaccharides of EPS and degrade biofilm structures [185,186].

Recent reports provide evidences of cathelicidin family of AMP, LL-37 as an anti-biofilm agent against polymicrobial biofilms. These peptides possess broad-spectrum anti-microbial and anti-biofilm properties. The role of LL-37 in the treatment of non-healing wounds comprising polymicrobial biofilms is described below [187]. This peptide showed an anti-biofilm effect against a wound infected with polymicrobial biofilms of *P. aeruginosa* and *S. aureus*. The LL-37 mechanism towards biofilms of *P. aeruginosa* was studied to discover how it prevents the twitching motility of bacteria, inhibits initial bacterial attachment and cell membrane perturbation, and inhibits QS signaling. In addition, a previous study showed inhibition of *P. aeruginosa* and *S. aureus* biofilms by LL-37 at a lower inhibitory concentration than that required to prevent sessile bacterial growth [188,189].

In another study, peptides showed anti-biofilm activity against the polymicrobial biofilms developed on root canal surfaces causing persistent apical periodontitis. A cationic peptide, human β-defensin-3, was reported with anti-bacterial and anti-inflammatory properties. In the current work, defensin-3 peptide inhibited the biofilm formation by *S. mutans*, *Lactobacillus salivarius*, *Actinomyces naeslundii*, and *E. faecalis* in an *in vitro* study. The anti-bacterial activity of β-defensin-3 is corroborated by its ability to form ionic interactions with a cell membrane, leading to membrane permeabilization. Some of the literature supports the biofilm disruption and inhibition properties of β-defensin-3 [190].

### 8.4. Quorum Sensing Inhibitors/Natural Products Based Anti-Biofilm Agents

Biofilm formation in some bacterial and fungal pathogens is controlled by a cell-to-cell signaling/QS mechanism through which specific autoinducer chemicals are released by the pathogen. The autoinducer at a threshold concentration interacts with their cognate receptor, which leads to the expression of genes essential for the host pathogenicity [191]. Innovative approaches targeting quorum sensing circuits showed potential anti-biofilm properties in several pathogens. Thus, novel anti-quorum sensing agents are recommended for anti-microbial therapeutics to manage chronic biofilm infections [192].

Studies showed that natural molecules are able to interfere with the bacterial/fungal signaling mechanism and thus inhibit polymicrobial infections. Volatile plant extracts, especially essential oils, are found to be anti-bacterial, anti-fungal and anti-cancer agents [193]. Pekmezovic et al. 2021 studied anti-quorum sensing activity of essential oils from citrus fruits, pompia and grapes against polymicrobial communities of *P. aeruginosa* and pathogenic fungi, *A. fumigatus* or *Scedosporium apiospermum*. Essential oils of both fruits affected quorum sensing in bacteria and inhibited polymicrobial biofilm formation [162]. In another study, a quorum sensing inhibitor was used to improve the activity of tobramycin antibiotic and finally inhibited the polymicrobial biofilms of *P. aeruginosa* and *S. aureus*. Quinazolinone, a quinolone QS circuit inhibitor, was used as adjuvant along with aminoglycoside antibiotic to eradicate chronic mixed-species biofilms [194].

Fatty acids are recognized as good anti-microbial agent against a variety of microorganisms. In addition, many reports provide evidence of fatty acids as anti-virulent and anti-biofilm agents against pathogens [195,196]. It has been observed that many saturated and unsaturated fatty acids affect biofilm formation by Gram-positive and Gram-negative bacteria and also fungi at lower concentrations. Fatty acids disrupted the biofilms of *P. aeruginosa*, *S. aureus*, *Serratia marcescens*, *Vibrio* sp., *B. cenocepacia* and *C. albicans* by targeting the adhesion, flagella and pili-mediated motility and virulence mechanisms regulated by the QS signaling circuits [197]. Interestingly, dual and three-species biofilm formations by *S. aureus*, *E. coli* O157:H7 and *C. albicans* were inhibited significantly by saw palmetto oil. The two naturally occurring fatty acids, lauric acid and myristic acid present in the palmetto oil, were reported as anti-biofilm agents against polymicrobial communities. As palmetto oil is one of the available health supplements in the market, the fatty acid components of oil can be used as safe anti-biofilm agents to prevent polymicrobial biofilm infections [196].

### 8.5. Phage Therapy

Phage therapy is a widely studied method to eradicate biofilms. Bacteriophages are viruses that are able to grow and replicate within bacterial cells and finally lead to the lysis of bacteria to release new virions [7]. Some phage genomes encode an EPS depolymerase enzyme capable of degrading biofilm matrices, thus inhibiting biofilm formation [198,199]. Application of phages is a safe, cheap and effective method of anti-biofilm therapy [200]. Single phage suspensions or phage cocktails are used generally in the anti-biofilm strategies. In a study, a novel phage was screened for anti-*P. aeruginosa* and anti-*Proteus mirabilis* properties, and the phage cocktail reduced the mixed-species biofilm formation on a urinary catheter model [201].

Phage therapy was employed by Chhibber and co-workers to mitigate the polymicrobial biofilms of *K. pneumoniae* B5055 and *P. aeruginosa* PAO. In the bacteriophage therapy, the phage KPO1K2 encoding depolymerase disrupted the biofilm matrix of *K. pneumoniae*, thus allowing the phage Pa29 access to *P. aeruginosa* biofilms. Pa29 is a phage without depolymerase activity; thus, synergistic combination of an anti-biofilm agent xylitol (a natural sugar alcohol) inhibited the polymicrobial biofilms [202]. Some phage-antibiotic combinations alleviated the infections caused by mixed-species biofilms [203]. A newly isolated anti-*P. aeruginosa* phage named the Pakpunavirus phage vB_PaM_EPA1, along with different classes of antibiotics, gentamicin, kanamycin, tetracycline, chloramphenicol, erythromycin, ciprofloxacin, and meropenem, exhibited biofilm inhibitory activity. The polymicrobial biofilms of *P. aeruginosa* and *S. aureus* often fail to eradicate by commonly available antibiotics owing to their resistance. Thus, the combination of phages that can penetrate the biofilms makes them more vulnerable to antibiotics and are considered as the best remedy to treat polymicrobial infections [160].

### 8.6. Probiotic Combinations

Recent studies suggest probiotic combinations as promising therapeutic method against polymicrobial biofilms. The most commonly used probiotic *Lactobacilli* sp. inhibited biofilms of several bacterial and fungal species [204,205]. Tan and colleagues studied the effect of a cell-free supernatant of probiotic bacterium, *L. rhamnosus,* on mixed bacterial-fungal interactions; the authors reported that a probiotic bacterial supernatant inhibited mixed biofilms formed by *C. albicans*, *C. tropicalis*, *S. epidermidis*, *S. salivarius*, *Rothia dentocariosa* [206]. In another study, combinations of different probiotic strains, *Saccharomyces boulardii*, *L. acidophilus*, *Bifidobacterium breve*, and *L. rhamnosus,* inhibited polymicrobial biofilms of bacteria-fungi. The probiotic filtrate inhibited the biofilms formed by polymicrobial communities of *C*. *albicans*, *C. tropicalis*, *E. coli* and *S. marcescens*. The study revealed the ability of probiotic strains to treat gastrointestinal-polymicrobial biofilm-associated infections [166]. Alpha-hemolytic streptococci, *S. salivarius* and *S. oralis,* are the two initial colonizers of the upper respiratory tract of healthy humans. These bacteria are known to protect epithelial cells from pathogenic invasions. In a study, the potential of *S. salivarius* and *S. oralis* as a probiotic strain against polymicrobial biofilm interactions was studied. The study concluded that α-hemolytic Streptococci isolated from a commercial product showed probiotic activity and inhibited mixed biofilm formation by pathogens of the upper respiratory tract, such as *S. pyogenes*, *S. pneumoniae*, *M. catarrhalis*, *S. aureus*, *S. epidermidis,* and *P. acnes* [207].

## 9. Conclusions

Biofilms are populations of microorganisms which produce exopolymer substances to protect from external environments. The interspecies or intraspecies interactions among different microorganisms help in the formation of polymicrobial biofilms. Polymicrobial biofilms exhibits competitive, synergistic or commensal interactions in their niches. The specialized polymicrobial interactions play significant roles in the pathogenesis of acute and chronic infections. Antibiotics have lost their efficacy towards mono-species biofilms and biofilms observed in polymicrobial interactions. In recent years, the emergence of large numbers of antibiotic-resistant strains among polymicrobial biofilms has increased the severity of polymicrobial infections. Insights into the *in vitro* and *in vivo* biofilm models of polymicrobial infections help to understand microbial diversity and to understand the nature of interactions. At present, novel and effective anti-microbial therapeutics with an ability to eradicate the polymicrobial biofilms of different microorganisms is highly required. We further discussed the developments of anti-microbial research to mitigate the bacterial-bacterial, bacterial-fungal and fungal-fungal interactions and the associated infections. In this review, the authors emphasize the application of culture-independent approaches, specifically metagenomics, in the identification and prevention of polymicrobial biofilms.

## Figures and Tables

**Figure 1 antibiotics-11-01731-f001:**
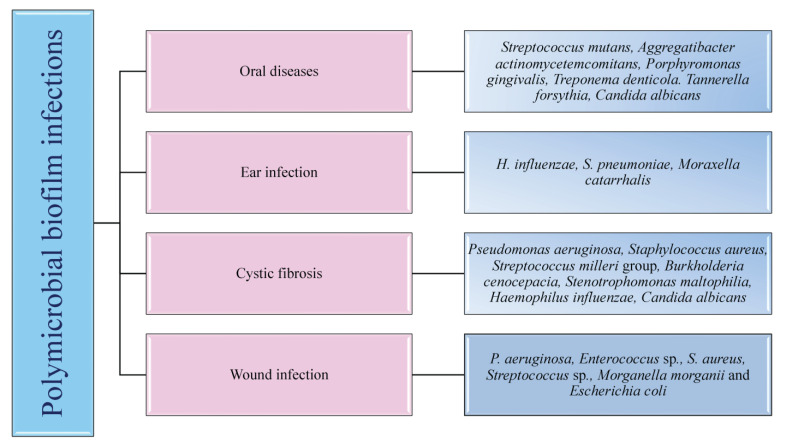
Different polymicrobial interactions and associated infections. The figure describes the bacterial-bacterial or bacterial-fungal interactions involved in the pathogenesis of chronic microbial infections.

**Figure 2 antibiotics-11-01731-f002:**
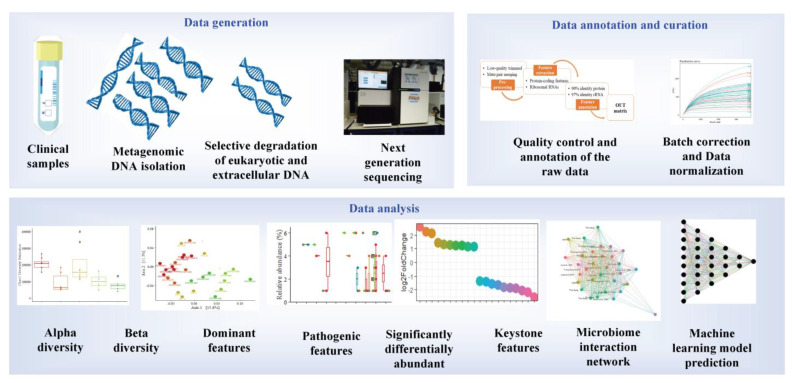
Diagnosis of polymicrobial infections through metagenomics involves multiple steps, which can be broadly divided into three sections-(i) data generation that involves sampling, DNA isolation, and extraction, (ii) quality control, annotation, and normalization to avoid biases due to sequencing depth or batches, and (iii) analysis of the data in detection of pathogens, virulence factors, antibiotic resistance, and finding patterns.

**Figure 3 antibiotics-11-01731-f003:**
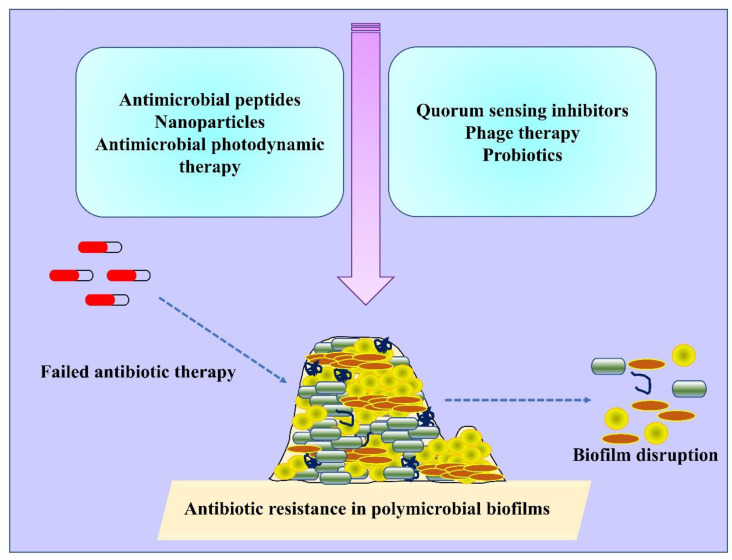
Illustration of different therapeutic approaches available so far to combat polymicrobial biofilm-associated infections.

**Figure 4 antibiotics-11-01731-f004:**
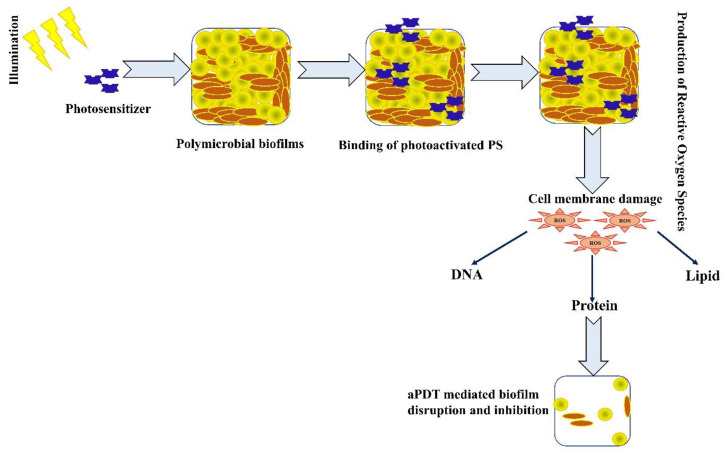
Photodynamic therapy–mediated killing of polymicrobial biofilms. The figure represents the mechanism of action of aPDT on biofilms where ROS targets the cell membrane and cellular components such as DNA, proteins and lipids.

**Table 1 antibiotics-11-01731-t001:** The *in vitro* and *in vivo* biofilm models used to study polymicrobial biofilms responsible for different chronic infections.

Infections	Microorganisms	*In Vitro*/*In Vivo*/Model Systems	References
Skin infections by *Staphylococcus aureus* and *Pseudomonas aeruginosa*	Commensal, *Staphylococcus epidermidis* and *Micrococcus luteus* and pathogenic *Staphylococcus aureus* and *Pseudomonas aeruginosa*	Immortalized keratinocytes (HaCat cells)	[102]
Chronic wound infections	MRSA, vancomycin-resistant *Enterococcus faecalis* (VRE) and *P. aeruginosa*	Lubbock Chronic Wound Biofilm (LCWB) model	[103]
Wound infections	Complex polymicrobial biofilms containing *Candida albicans*, *P. aeruginosa*, *S. aureus*, *Staphylococcus hominis*, *Corynebacterium simulans*, *Streptococcus agalactiae*, *Finegoldia magna*, *Prevotella buccalis*, *Porphyromonas asaccharolytica*, *Anaerococcus vaginalis*, and *Peptoniphilus gorbachii*	Skin epidermis model	[104]
Observed in the lungs of cystic fibrosis patients	*C. albicans* and *P. aeruginosa*	*Caenorhabditis elegans* (Nematode)	[105]
Chronic wound infections	*S. aureus*, *P. aeruginosa*, *E. faecalis* and *Finegoldia magna*	Mice	[57]
Diabetes-associated manifestations such as lower-limb amputations due to wound infections	*Escherichia coli*, *Bacteroides fragilis*, and *Clostridium perfringens*	Human type 2 diabetes model of mice	[106]
Periodontal disease	*Porphyromonas gingivalis* and *Streptococcus gordonii*	Murine model of periodontitis	[107]
Chronic periodontitis	*P. gingivalis* and *Treponema denticola*	Murine model of periodontitis	[108]
Otitis media	*Haemophilus influenzae* and *Moraxella catarrhalis*	Chinchilla infection model of otitis media	[109]
Acute otitis media	*Moraxella catarrhalis*, *Streptococcus pneumoniae* and non-typeable *H. influenza*	*In vitro* nasopharyngeal colonization model	[32]

**Table 2 antibiotics-11-01731-t002:** Various strategies have been devised to treat polymicrobial biofilm. The biofilm disruption is mainly mediated by the loss of biomass, reduced cell adhesion, and interference in the biofilm matrix structure.

Strategies	Polymicrobial Biofilm	Mechanism	References
**Curcumin loaded with chitosan nanoparticle**	*Candida albicans* and *Staphylococcus aureus*	Biofilm disruption	[153]
**AgNP functionalised silicone elastomer**	*C. albicans* and MRSA	Biofilm inhibition	[154]
**Pentadecanoic acid coated on** **polydimethylsiloxane (PDMS) surface**	*C. albicans-Klebsiella pneumoniae*	Polymicrobial biofilm prevention	[155]
**Electrospun membranes of poly (lactic acid) and carvacrol**	*C. albicans* and *S. aureus*	Decrease in the CFUs, biomass, and metabolic activity of 24- and 48-h biofilms in both single and mixed biofilms	[156]
**Gh625-GCGKKK Peptide**	*Candida tropicalis*–*Serratia marcescens* and *C. tropicalis*–*S. aureus*	Reduced biofilm architecture, interfering cell adhesion, prevention of long-term formation of polymicrobial biofilm on silicone surface	[157]
**Synthetic cationic AMP, Nal-P-113**	*Streptococcus gordonii*, *Fusobacterium nucleatum*, *Porphyromonas gingivalis*,	Bactericidal activity in both planktonic and polymicrobial biofilm states	[158]
**Cholic acid peptide conjugates (CAPs)**	*C. albicans* and *S. aureus*	Reduces interkingdom polymicrobial biofilm formation and also active towards persister cells as well as stationery cells	[159]
**Lytic phage, EPA1 with antibiotics such as (gentamicin, kanamycin, tetracycline, chloramphenicol, erythromycin, ciprofloxacin, and meropenem)**	*P. aeruginosa* and *S. aureus*	Mono and dual species biofilm Inhibition	[160]
**Phages, PYO and Sb-1 with Ciprofloxacin**	*P. aeruginosa* and *S. aureus*	Biofilm matrix inhibition	[161]
**Pompia and grapefruit essential oils**	*P. aeruginosa*, *Aspergillus fumigatus* or *Scedosporium apiospermum*	Mono and Polymicrobial biofilm inhibition	[162]
**Oxantel**	*Treponema denticola*, *Porphyromonas gingivalis* and *Tannerella forsythia*.	Polymicrobial biofilm disruption	[163]
**Antimicrobial Photodynamic therapy- Chlorin e6**	*H. influenzae*, *S. pneumoniae* and *M. catarrhalis*	Biofilm disruption	[164]
**Zn(II)chlorin e6 methyl ester (Zn(II)e_6_Me)**	*Enterococcus faecalis* and *C. albicans*	Loss of biofilm biomass	[165]
**Probiotics, *Saccharomyces boulardii*, *Lactobacillus acidophilus*, *Bifidobacterium breve*, and *Lactobacillus rhamnosus***	*C. albicans* or *C. tropicalis* combined with *E. coli* and *S. marcescens*	Inhibition of candidal pathogenic determinants, prevent adhesion and biofilm formation	[166]
**Glycoside hydrolases, α-amylase and cellulase**	*P. aeruginosa* and *S. aureus*	Breakdown of complex sugars and disruption of mono and coculture biofilm	[65]

## Data Availability

Not applicable.

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
