# Peer review of "Polymicrobial Infections and Biofilms: Clinical Significance and Eradication Strategies"

_antibiotics, 2022, doi:10.3390/antibiotics11121731_

Round 1

Reviewer 1 Report

Manuscript  title : Polymicrobial biofilms: Clinical significance and eradication  strategies is not correlated with body of manuscript. It hard to find out major aims  of presented  review.

The basic question is that Authors mixed –up bacterial population interaction with bacterial biofilms.

How Authors defined multi-species biofilms – is that fungal/bacterial biomass growing on solid  surfaces? The major concern is that review ae not focus and too many topics resulted  on selected , basic information provided. It is suggested to focus on one type of biofilms –i.e. oral and more detailed describe it . Indeed polymicrobial biofilms are important problem but presented manuscript do not add deeply structuralized information about such often studied matter.

On present form manuscript cannot be published.

Author Response

Response to the reviewer is attached.

Reviewer 2 Report

This review of Anju et al is extensive, well written and well discussed. I only have some minor suggestions:

I would add at the end of the title: "based on metagenomic approaches"

Introduction: line 33: "bacteria are observed as biofilms rather than planktonic forms".

Introduction: lines 75-79: Check the word size of some species.

Introduction: line 103: Write Streptococcus in S. mutans, as it is the first time cited in the text. Then in line 141 it can be shortened to S. mutans. 

Author Response

Response to the reviewer is attached.

Reviewer 3 Report

The study “Polymicrobial biofilms: Clinical significance and eradication strategies” form Anju et al. explores an interesting review article. But I order in order to increase the usefulness and significance of the study, it needs a revision before being considered suitable for readers and there are some points to overcome for acceptance.

In this review article, author focussed on the polymicrobial interactions among bacterial-bacterial, bacterial-fungal and fungal-fungal aggregations based on in vitro and in vivo models and also different therapeutic interventions available for polymicrobial biofilms. However, the introduction of these sections is quite plain, not in deep. It is recommended to tone up the introduction section.

Line 37 need an appropriate reference.

Recommended to increase font size of microbe’s names in figure 1.

Picture quality is poor in figure 2. X or Y axis scale not visible.

Line 44 and 45 need an appropriate reference.

In section 2 (line 88) to many unwanted paragraphs. Please rearrange it carefully.

Recommended to represent the regulating approaches for polymicrobial biofilm in tabular manner. (E.g. Strategy, Mechanism, and respective reference/s).

Recommended to represent main phases leading to the development and dispersal of polymicrobial biofilm in introduction section.

It is suggested a moderate English revision by an English native speaker in order to polish text from typos and imperfections.

Unwanted spacing and typo mistakes throughout the manuscript. Need to be check and correct carefully.

Double check the way of adding references in the main text body and reference section as per journal guidelines.

Please check the numbering of bibliography.

Author Response

Response to the reviewer is attached.

Round 2

Reviewer 1 Report

The major concern is that review ae not focus and too many topics resulted on selected , basic information provided. It is suggested to focus on one type of biofilms –i.e. oral and more detailed describe it . Indeed polymicrobial biofilms are important problem but presented manuscript do not add deeply structuralized information about such often studied matter. On present form manuscript cannot be published.

Authors do not repond to major question - see above 

Reviewer 3 Report

The manuscript is thoroughly revised and the authors have addressed all the comments meticulously. However, the manuscript requires thorough proofreading by a native speaker.